# Evaluation of Cardiovascular Pharmacotherapy Guideline Adherence and Risk Factor Control in Portuguese Community Pharmacy Patients

**DOI:** 10.3390/ijerph19106170

**Published:** 2022-05-19

**Authors:** Anabela Fonseca, Tácio de Mendonça Lima, Fernando Fernandez-Llimos, Maria Margarida Castel-Branco, Isabel Vitória Figueiredo

**Affiliations:** 1Pharmacology and Pharmaceutical Care Laboratory, Faculty of Pharmacy, University Coimbra, 3000-548 Coimbra, Portugal; anabelafon@gmail.com (A.F.); mmcb@ci.uc.pt (M.M.C.-B.); isabel@ff.uc.pt (I.V.F.); 2Department of Pharmaceutical Sciences, Federal Rural University of Rio de Janeiro, Seropédica 23897-090, RJ, Brazil; 3Laboratory of Pharmacology, Department of Drug Sciences, Faculty of Pharmacy, University of Porto, 4050-313 Porto, Portugal; fllimos@ff.up.pt; 4Coimbra Institute for Clinical and Biomedical Research (iCBR), 3000-548 Coimbra, Portugal

**Keywords:** cardiovascular diseases, risk assessment, guideline adherence, pharmacists, dyslipidemias, hypertension, diabetes mellitus, cross-sectional studies, Portugal

## Abstract

Background: Cardiovascular disease (CVD) remains the leading cause of death worldwide. Assessing the patients’ CVD risk, controlling the risk factors, and ensuring the guideline-adherent cardiovascular pharmacotherapy are crucial interventions to improve health outcomes. This study aimed to evaluate the potential of pharmacists to improve the adherence to pharmacotherapy guidelines and the achievement of risk factor goals among patients who attended a community pharmacy. Methods: We conducted a single-center cross-sectional study. We performed in-pharmacy point-of-care testing, blood pressure and anthropometric measurements, and reviewed patients’ pharmacotherapy, based on European Society of Cardiology guidelines. Results: Of the 333 patients, 63.1% were in the high/very high risk category, 91.9% showed at least two modifiable risk factors, and in 61.9% of patients the cardiovascular pharmacotherapy was non-adherent to the current guidelines, failing to reach treatment goals. The lipid-lowering therapy was the least guideline adherent, with a suboptimal use of statins. However, we found no statistically significant difference between the guideline-adherent and the non-adherent group in terms of risk factor control. The pharmacist recommended 603 interventions to adhere to the guidelines. Conclusions: Community pharmacists are able to identify opportunities to optimize cardiovascular pharmacotherapy and support the patients to achieve cardiovascular risk factor goals, based on evidence-based guidelines, contributing to the improvement of CVD management.

## 1. Introduction

Cardiovascular disease (CVD) remains the leading cause of death worldwide, as 17.79 million people died from CVD in 2017 [1]. Assessing the global CVD risk in the individuals and controlling their CVD risk factors, such as dyslipidemia, hypertension, diabetes, tobacco use, obesity, physical inactivity, harmful alcohol use, and unhealthy diet, is important to adequately manage patients and cardiovascular events [2]. Cardiovascular pharmacotherapy plays a major role in the management of CVD and CVD risk factors and has proven to be the most beneficial intervention [3]. However, although being on cardiovascular pharmacotherapy, many patients do not achieve their risk factor goals, showing that CVD risk is not adequately addressed. Adherence to cardiovascular pharmacotherapy guidelines is associated with improved outcomes in primary and secondary cardiovascular prevention and reduces CVD [4,5,6].

Many studies have supported the involvement of pharmacists as healthcare providers in managing patients with hypertension, diabetes, and dyslipidemia and in optimizing CVD outcomes, by detecting uncontrolled cardiovascular risk factors, suboptimal and non-guideline-adherent pharmacotherapy [7,8], and by performing medication review to optimize drug use [9,10].

The increasing prevalence of chronic conditions in Portugal has profound consequences on the national healthcare service, requiring a shift in the current healthcare model. The Portuguese community pharmacist’s scope of practice was expanded in 2018 to include new services routinely provided and enforced by law: nutrition appointments; therapy adherence programs, medicine reconciliation, services utilizing multicompartment aids, and education programs on the use of medical devices; performance of rapid tests for HIV, HCV, and HBV screening (point-of-care tests), including pre-and post-test counseling and referral of positive cases to hospital care; and basic nursing services [11]. Pharmacies may also promote campaigns and programs for health literacy, disease prevention, and healthy lifestyle promotion. Most Portuguese pharmacies perform point-of-care tests that enable CVD risk evaluation. These services are freely priced by pharmacies and paid out of pocket by users, but there are no publicly available data on the number of services or pricing [12]. We have already demonstrated that community pharmacists play a relevant role in providing cardiovascular risk screening and detecting CVD risk factors and at-risk customers [13]. Currently, limited research on the use of guideline-based cardiovascular therapy exists in Portugal. Thus, the objective of this study was to evaluate the opportunities pharmacists have in detecting non-adherence to cardiovascular pharmacotherapy guidelines and uncontrolled risk factors in community pharmacy patients and to demonstrate the role pharmacists can play in improving the quality of care through optimization of cardiovascular pharmacotherapy.

## 2. Materials and Methods

### 2.1. Study Design and Data Collection

A single-center, cross-sectional study was conducted to evaluate the cardiovascular pharmacotherapy guideline adherence and risk factor control. The enrollment process occurred in a community pharmacy in a central Portuguese city. All the customers who entered the pharmacy during a 70-h working week (Monday through Saturday) were registered with name, date of birth, and telephone number. Then, the pharmacist contacted each pharmacy customer over the phone and made an invitation to participate in the study. The pharmacist performed a CVD risk assessment for the participants who agreed to participate in the study in a dedicated counseling room, where privacy could be maintained within the pharmacy with no interference by routine pharmacy processes, for seven months.

The inclusion criteria for this study were customers who were on cardiovascular pharmacotherapy. The customers who were not on cardiovascular pharmacotherapy were included in a screening program [13]. The exclusion criteria were customers younger than 18 years, pregnant customers, or mentally disabled persons who were unable to consent. To avoid any selection bias, the pharmacist contacted each customer and used a standard invitation speech in every phone call, clarified possible doubts, scheduled an appointment for data and sample collection, and informed the customer about the requirement to maintain a 12-h fasting and bring all medications. All participants gave their written informed consent before the interview was initiated, and they had the opportunity to raise doubts or questions before their assessment.

The sociodemographic characteristics and health data were self-reported and ascertained via questionnaires.

The pharmacist collected capillary blood for the analysis of blood glucose level, HbA1c, total cholesterol (TC), high-density lipoprotein (HDL), and triglycerides (TG) and calculated low-density lipoprotein (LDL) for in-pharmacy point-of-care testing, obtaining the results in a 15-min workflow. The pharmacist also performed a physical examination comprising an evaluation of the systolic and diastolic blood pressure (BP), heart rate, weight, height, and abdominal perimeter at waist level. In the community pharmacy, these evaluations were performed by a licensed pharmacist. The devices and the training of the investigator pharmacist on equipment operation were provided by World Care & Diagnostics. The blood glucose levels were measured with an Accu-Chek Performa device (Roche, Basel, Switzerland); lipids were measured from whole blood with a Cobas b101 system (Roche, Basel, Switzerland). The physical examination was performed with a Tensoval Duo Control (Hartmann, Heidenheim, Germany), an electronic stadiometer (Exclusivas Iglesias, Cangas-PO, Spain), and an anthropometric measuring tape.

Based on the obtained results, the pharmacist performed the CVD risk assessment, by applying the Systematic Coronary Risk Evaluation (SCORE) model [14,15,16].

Regarding smoking habits, only current smokers were considered as presenting this major modifiable CVD risk factor. Overweight patients presenting a body mass index (BMI) ≥ 25 kg/m^2^ and obese patients presenting BMI ≥ 30 kg/m^2^ were considered at-risk [17]. The European Society of Cardiology (ESC) recommends that healthy adults of all ages perform at least 150 min a week of moderate-intensity, 75 min a week of vigorous-intensity aerobic physical activity, or an equivalent combination thereof [17]. Less physical activity than recommended was considered as presenting sedentary behavior.

Fasting blood glucose was classified into normal blood glucose (<100 mg/dL), impaired glucose tolerance (100–125 mg/dL), and diabetes (≥126 mg/dL) and HbA1c ≥ 6.5%. The HbA1c targets were determined according to the recommendations of the 2019 ESC Guidelines on diabetes, pre-diabetes, and CVD [15].

We considered the 2018 ESC/ESH Guidelines for the management of arterial hypertension, blood pressure thresholds for treatment initiation, and treatment targets in hypertensive patients [18].

The dyslipidemia risk factor was considered positive when the patient presented: LDL-C > 55 mg/dL for very high-risk patients, LDL-C > 70 mg/dL for high-risk patients, LDL-C > 100 mg/dL for moderate-risk patients, and LDL-C > 116 mg/dL for low-risk patients or secondary lipid parameter non-HDL-C > 85,100, and 130 mg/dL for very-high-, high-, and moderate-risk patients, respectively, a total cholesterol > 190 mg/dL, or triglycerides >150 mg/dL [16].

For the evaluation of cardiovascular pharmacotherapy guideline adherence, the pharmacist performed a type 2a medication review [19] and used the WHO Anatomical Therapeutic Chemical Classification System until the fourth level of the code to register the medication. The lipid-lowering therapy with statins was classified into three different dosage intensity categories, high-, moderate-, and low-intensity statin, according to the American College of Cardiology and the American Heart Association guidelines [20]. In order to evaluate the ESC Guidelines adherence [21], we focused on the treatment of the main cardiovascular risk factors, namely type 2 diabetes, dyslipidemia, and hypertension [15,16,18]. For this purpose, we established quality indicators (QI), which enabled the quantification of adherence to guideline recommendations. For guideline adherence in patients with hypertension, we considered the recommendation of the inclusion of an angiotensin-converting enzyme inhibitor (ACEI), or an angiotensin receptor blocker (ARB) in patients who are intolerant to ACEI (QI-1); and that blood pressure control often requires multiple drug therapy with a renin-angiotensin-aldosterone system (RAAS) inhibitor, a calcium channel blocker (CCB), and diuretics (QI-2). Regarding guideline adherence in patients with type 2 diabetes, we considered that metformin is recommended as first-line therapy in patients without previous atherosclerotic CVD, chronic kidney disease, or heart failure and should be considered in persons with type 2 diabetes and atherosclerotic CVD unless contraindications are present (QI-3); and that in patients with type 2 diabetes and atherosclerotic CVD or in those who are at very high/high cardiovascular risk, the use of glucagon-like peptide-1 receptor agonist (GLP-1RA) or a sodium-glucose cotransporter-2 (SGLT2) inhibitor is recommended to reduce cardiovascular and cardiorenal outcomes (QI-4). Regarding guideline adherence in patients with dyslipidemia, we considered the recommendation that a high-intensity statin should be prescribed up to the highest tolerated dose to reach the LDL-C goals set for the specific risk category, and if the goals are not achieved with the maximum tolerated dose of a statin, a combination with ezetimibe is recommended (QI-5). In addition, we considered the recommendation that all smoking of tobacco should be stopped, and that in smokers, offering follow-up support, nicotine replacement therapy, varenicline, and bupropion individually or in combination should be considered (QI-6). Moreover, and as antiplatelet drugs are the cornerstone of secondary cardiovascular prevention, we considered that aspirin 75 to 100 mg daily is recommended for secondary prevention of CVD, or clopidogrel 75 mg daily in case of aspirin intolerance (QI-7); in patients with diabetes mellitus (DM) at high/very high risk, aspirin may be considered in primary prevention (QI-8); and that antiplatelet therapy is not recommended in individuals with low/moderate cardiovascular risk due to the increased risk of major bleeding (QI-9). The QI were based on A level treatment evidence, except QI-3, which was based on B level evidence, and a strong recommendation class, except QI-8, where the treatment may be considered.

### 2.2. Data Analysis

We used descriptive statistics to explore the characteristics of the study population and chi-square to analyze possible associations between variables. For the data analysis, we used the SPSS v.24 (IBM, Armonk, NY, USA) and considered a *p*-value lower than 0.05 as statistically significant. For each QI we analyzed what proportion of the patients had been treated adherent and not adherent to guideline recommendations and the degree of control of the risk factors in each group. To analyze possible associations between the main quality indicators of guideline adherence and CVD risk factors, we used chi-squared.

### 2.3. Ethical Statement

Ethics approval for conducting this study was received from the Ethics Committee of the Faculty of Medicine of Coimbra University in March 2017 (reference number: CE_Proc. CE-011/2017). Consent to publish was obtained from the patients as indicated in the Consent for Participation Form, which was part of the ethics application forms submitted to the Coimbra University Institutional Review Board. The customers gave their consent to register their name, date of birth, and telephone number, to the pharmacist/pharmacy technician/trainee, at the enrollment process, signing the digital consent form available on the informatic system Sifarma 2000 (Glintt, Lisbon, Portugal) developed by the Portuguese pharmacy association (ANF—Associação Nacional das Farmácias).

## 3. Results

During the study period, 1261 customers entered the pharmacy. We were able to contact 1101 of them and 513 did not accept to participate or did not attend the scheduled appointment, 255 did not meet the inclusion criteria, because they were not taking cardiovascular pharmacotherapy, and 333 met the inclusion criteria and were included into the study.

The descriptive characteristics of the patients and access to healthcare are shown in Table 1. The mean time spent with each patient in the interview was 27.1 min (SD = 6.40), with a minimum of 14 and a maximum of 55 min. The patients had a mean age of 65.00 years (SD = 11.23). Of the 333 patients, 84.7% (*n* = 282) were regular customers. The patients visited the pharmacy about four times more often than the physician.

### 3.1. Cardiovascular Risk and Risk Factor Assessment

We found that 63.1% patients (*n* = 210) were classified into a high or very-high cardiovascular risk category. According to the SCORE risk evaluation, of the 333 patients, 8.1% (*n* = 27) were low-risk patients, 28.8% (*n* = 96) were moderate-risk patients, 16.2% (*n* = 54) were high-risk patients, and 46.9% (*n* = 156) were very high-risk patients. We considered the patients < 40 years (*n* = 6) as low-risk patients, since none of them reached a relative SCORE >3. The modifiable cardiovascular risk factors of the analyzed patients are shown in Table 2.

Regarding major modifiable risk factors, 0.9% (*n* = 3) patients presented no risk factors, 27.6% (*n* = 92) patients presented one or two risk factors, and 71.5% (*n* = 238) patients presented three or more risk factors. Thus, 99.1% (*n* = 330) had at least one modifiable CVD risk factor. Furthermore, the mean number of CVD risk factors increases as the CVD risk category increases: 2.37 (SD = 1.01) in low-risk patients, 2.69 (SD = 1.00) in moderate-risk patients, 3.02 (SD = 1.01) in high-risk patients, and 3.31 (SD = 1.01) in very high-risk patients. Apart from these CVD risk factors and after excluding the patients with diagnosed type 2 and type 1 diabetes (*n* = 70), among the remaining 263 patients, 33.1% (*n* = 87) patients had fasting glucose levels 102–125 mg/dL, 3.0% (*n* = 8) patients had fasting glucose levels ≥ 126 mg/dL, and 3.8% (*n* = 10) patients had HbA1c ≥ 6.5%.

### 3.2. Evaluation of Cardiovascular Pharmacotherapy Guideline Adherence

The 333 patients were taking 821 prescription medications, comprising 995 drugs, considering the combinations of drugs in single-pill combination therapy, which resulted in a mean of 2.47 (SD = 1.63) drugs per patient. The most prescribed medication were antihypertensives (*n* = 399) (diuretics (*n* = 139), ARB (*n* = 112), ACEI (*n* = 78), beta-blocking agents (*n* = 70)), statins (*n* = 196), antithrombotic agents (*n* = 109), and metformin (*n* = 60). The most frequently used statin was atorvastatin 41.3% (*n* = 81) and simvastatin 39.8% (*n* = 78). In terms of statin intensity, 6.1% (*n* = 12) patients were taking low-intensity statins, 91.3% (*n* = 179) patients were taking moderate-intensity statins, and 2.6% (*n* = 5) patients were taking high-intensity statins. Of the patients treated with antihypertensives, 19.0% (*n* = 43) were treated with monotherapy, 75.3% (*n* = 171) were treated with two or three antihypertensive drugs, and 5.7% (*n* = 13) were treated with four or five antihypertensive drugs.

Of the 333 patients analyzed, 9.6% (*n* = 32) showed high BP but were not treated for hypertension; 29.1% (*n* = 97) showed high lipid levels but were not treated for dyslipidemia; and 3.0% (*n* = 10) showed high HbA1c but were not treated for type 2 diabetes; 68.8% (*n* = 229) were treated for hypertension, 60.7% (*n* = 202) for dyslipidemia, and 20.1% (*n* = 67) for type 2 diabetes; and from these, 23.6% (*n* = 54) reached blood pressure target, 38.1% (*n* = 77) reached lipid targets, and 74.6% (*n* = 50) reached HbA1c targets, respectively.

The results of the QI for guideline adherence for the main CVD risk are shown in Table 3.

None of the 26 smokers had been prescribed with follow-up support, nicotine replacement therapy, varenicline, or bupropion individually or in combination (QI-6). Of the 39 patients in secondary cardiovascular prevention, 71.8% (*n* = 28) were taking aspirin, as recommended by the guidelines (QI-7). Of the 60 patients with type 2 diabetes at high/very high risk in primary prevention, 33.3% (*n* = 20) were taking aspirin, as recommended by the guidelines (QI-8). Of the 123 patients with low/moderate cardiovascular risk, 92.7% (*n* = 114) were not on antiplatelet therapy, as recommended by the guidelines, due to the increased risk of major bleeding (QI-9).

Almost 60.0% (*n* = 120) patients treated for dyslipidemia were not reaching the LDL-C targets, but 98.3% (*n* = 118) of these patients were on moderate and low-intensity statins or fenofibrates.

In terms of BP control, 76.4% of the treated patients were not reaching their BP targets and from these, 28.7% (*n* = 38) were treated with monotherapy. Indeed, the higher the CVD risk, the more intensive the antihypertensive therapeutic approach found, in terms of the number of antihypertensive drugs: low-risk (*n* = 12) 1.58; moderate-risk (*n* = 54) 1.83; high-/very-high-risk (*n*=161) 1.90 drugs. This approach resulted in an increased rate of control, low-risk (8.3%), moderate-risk (20.4%), and high-risk (39.5%), except for the very high-risk category where the rate of control dropped (20.3%). We found within the group of patients in the high and very high CVD risk category, 16 patients without antihypertensive therapy and with grade 1 or 2 hypertension.

We found that 20.3% (*n* = 54) nondiabetic patients with metabolic syndrome were treated with beta-blockers or thiazide diuretics, despite these antihypertensive drug classes may affect diabetes onset, and 68.5% (*n* = 37) patients presented abnormal fasting glucose levels.

## 4. Discussion

We found low adherence to cardiovascular pharmacotherapy guidelines, a lack of treatment intensification, and poor risk factor control in Portuguese patients visiting a community pharmacy. In light of the current ESC guideline recommendations, we identified 61.9% (*n* = 206) treated patients, whose cardiovascular therapy was non-adherent with evidence-based guidelines, failing to reach target levels for CVD risk factors. In 333 patients we identified 603 opportunities for intervention to increase adherence to the guidelines, manage cardiovascular therapy, improve outcomes, and reduce CVD risk. The lipid-lowering therapy was found to be the least guideline adherent, with a suboptimal use of statins. Moreover, we found a high prevalence of CVD risk factors, as 91.9% (*n* = 306) of the analyzed patients showed at least two modifiable CVD risk factors.

### 4.1. Cardiovascular Risk and Risk Factor Assessment

The cardiovascular risk assessment conducted on the patients revealed a high prevalence of cardiovascular risk factors and a high cardiovascular risk status, as 91.9% presented at least two uncontrolled modifiable risk factors and most patients (63.1%) were classified into high to very-high cardiovascular risk category. The five most prevalent modifiable risk factors were overweight or obesity, dyslipidemia, hypertension, sedentary behavior, and abnormal fasting glucose levels. Our data show a poor risk factor management and a high incidence of at-risk patients, who are already being followed and treated by their physician, corroborating the findings of other national studies. The results of our study revealed that the treatment rates for hypertension (87.7%) are higher than most of the former national studies (PAP, 38.9%; PHYSA, 74.9%; INSEF, 69.4%; e_COR, 69.9%; Precise, 98.0%). However, the blood pressure control was lower with respect to former national studies, with only 23.6% of controlled patients (PAP 28.7%; PHYSA, 55.7%; INSEF, 71.3%; e_COR, 32.1%; Precise, 56.7%) [22,23,24,25,26]. Moreover, the current ESC guidelines recommend more strict blood pressure and lipid targets, which renders the control of hypertension and dyslipidemia an even more challenging task. Concerning LDL-C, we found lower treatment rates (60.7%) and lower control rates (38.1%) than in the e_COR study, with 71.4% patients treated and 52.1% controlled [25].

We found that 74.6% (*n* = 50) patients with type 2 diabetes had HbA1c within their targets, which were individualized according to the duration of DM and comorbidities. The control rates observed in the present study were higher than those found by a Portuguese health examination survey [27] and the e_COR study [25], with control rates of 63.2% and 64.0% respectively, but where HbA1c targets lower than 7.0% were generalized for all patients with type 2 diabetes, which in our study would also result in lower control rates (67.2%). Most of the patients with type 2 diabetes failed to comply with the LDL-C (74.6%) and blood pressure (68.7%) recommended clinical targets. These findings were similar to the e_COR study [25], where the authors found 71.9% and 59.0% patients with type 2 diabetes with uncontrolled LDL-C and high blood pressure, respectively. Moreover, 58.2% (*n* = 39) type 2 diabetes patients presented simultaneously a dyslipidaemia, hypertension, and obesity constellation.

One of the major differences that the ESC guidelines [15], introduced to the 2016 European Guidelines on CV disease prevention in clinical practice [17], were the cardiovascular risk categories classification in patients with diabetes, who are now considered to be at very high risk of CVD, when presenting three or more major risk factor (e.g., age, hypertension, dyslipidemia, smoking, obesity). While with the previous guidelines 49.3% (*n* = 33) patients in our study would be classified as being at high risk and 50.7% (*n* = 34) at very high risk, the new guidelines classify only 13.4% (*n* = 9) patients with type 2 diabetes at high-risk and 86.6% (*n* = 58) at very high risk for CVD. These changes showed the previous underestimation of the CVD risk in patients with type 2 diabetes presenting a collection of CVD risk factors. Another innovation of the new guidelines is the lower treatment targets for LDL-C across the cardiovascular risk categories in a stepwise approach.

The 2021 ESC Guidelines on CVD prevention in clinical practice were released in September 2021 [21]. In terms of target levels for the risk factors, the newest guidelines are in line with the previous guidelines on diabetes and pre-diabetes (2019) [15] on dyslipidemias [16], and on arterial hypertension (2018) [18]. Another relevant difference are the new SCORE2 and SCORE2-OP charts, which predict the 10-year risk of fatal and non-fatal cardiovascular events, while the previous charts predicted the 10-year risk of fatal cardiovascular events. These new charts were calibrated to four clusters of countries (low, moderate, high, and very high CVD risk) that were grouped based on national CVD mortality rates published by the World Health Organization. Based on these mortality rates, Portugal was considered as moderate risk region for CVD, which influenced the risk category stratification. Another difference compared to former guidelines is the stepwise approach to risk factor treatment and treatment intensification to reach risk factor goals. This approach is facilitated by the communication to patients of treatment benefits of risk factors in an understandable way, with charts showing the average lifetime benefit of smoking cessation, lipid-lowering, and BP-lowering, expressed as extra CVD-free life-years, which may improve the shared decision-making process.

### 4.2. Evaluation of Cardiovascular Pharmacotherapy Guideline Adherence

The analysis of the QI revealed many opportunities for improvement in the prescribed cardiovascular pharmacotherapy. The results of the adherence rate showed that the QI with the lowest degree of adherence was smoking cessation therapy (QI-6) because none of the smokers in our study were on follow-up support, nicotine replacement therapy, varenicline, or bupropion. Statin therapy (QI-5), SGLT2 inhibitors and GLP-1RAs (QI-4), and the multiple drug therapy in hypertension (QI-2) were the cardiovascular pharmacotherapy with the most opportunities for intervention, due to alarmingly low adherence with these level A evidence clinical recommendations. Thus, the lipid-lowering therapy was found to be the least guideline adherent, with a suboptimal use of statins in terms of intensity.

As stated by ESC guidelines, combination treatment is needed to control BP in most patients and the association of multiple pharmacological classes is frequently needed to improve BP control. Despite these ESC guideline recommendations, in the uncontrolled BP group, we found a high number of patients treated with monotherapy. The high prevalence of vascular complications and the role of high BP is the leading global contributor to CVD [28], which reinforces the need to improve BP management to minimize CVD risk.

Although SGLT2 inhibitors and GLP-1RAs are recommended in patients with type 2 diabetes and CVD, or at high/very high CVD risk, to reduce cardiovascular events, in our study only 6.0% (*n* = 4) of eligible patients are receiving these therapies, which is coincident with other studies [29,30].

The use of metformin in patients with type 2 diabetes without previous atherosclerotic CVD, CKD, HF, or with atherosclerotic CVD, unless contraindicated, showed a high adherence (87.7%) in our study, probably because of the long experience with metformin in clinical practice.

However, we found no statistically significant difference between the guideline-adherent and the non-adherent groups, in terms of risk factor control.

As in other countries [31], the use of guideline-based cardiovascular therapy in treated patients in Portugal is low. In this study, we identified 603 opportunities for intervention, to improve adherence to current ESC guidelines, improve therapeutic outcomes on blood pressure, lipid, and HbA1c targets, in patients with cardiovascular pharmacotherapy. In fact, in very-high cardiovascular risk patients, the probability to present non-controlled risk factors was higher, confirming the importance to pay special attention to those patients, despite being already on cardiovascular pharmacotherapy.

Some limitations of this study are related to the fact that Portuguese community pharmacists do not have access to the patients’ medical records. The pharmacist depended on point-of-care testing performed in the pharmacy and lab test results and the medication history provided by the patients during the interview. In this study, patients’ medication adherence was not evaluated with a systematic and validated method. This, together with other inappropriate behaviors, could contribute to poor outcomes in patients with guideline adherent prescriptions. Some data used in this study were retrospective and self-reported, with the potential risk of recall bias. Moreover, the study was conducted in a single pharmacy, which may limit the generalizability.

## 5. Conclusions

We identified a high cardiovascular risk factor prevalence and opportunities for intervention through medication review to optimize cardiovascular pharmacotherapy, based on defined QIs and supported by evidence-based clinical recommendations. However, guideline adherence per se did not show improved risk factor control, proving that cardiovascular disease management requires a multifactorial approach. This study may provide the groundwork for further, larger, multicentric studies to prove the positive impact of pharmacist-led medication review on CVD.

## Figures and Tables

**Table 1 ijerph-19-06170-t001:** Descriptive characteristics and healthcare access of the sample of the patients (*n* = 333).

Characteristic	Description	*n*	%
Gender	Male	164	49.2
	Female	169	50.8
Age	18–49 years	27	8.1
	50–64 years	130	39.0
	65–79 years	147	44.2
	>80 years	29	8.7
Level of education	Illiterate 0 years	13	3.9
	1–4 years	191	57.4
	5–6 years	38	11.4
	7–9 years	34	10.2
	10–12 years	31	9.3
	University degree or Master’s degree	25	7.5
	PhD	1	0.3
Professional situation	Employed	99	29.7
	Unemployed	18	5.4
	Retired	193	58.0
	Student	1	0.3
	Domestic	22	6.6
Attributed primary care physician	Yes	325	97.6
	No, or do not know	8	2.4
Hospitalization last year	Yes	54	16.2
Resorted to the emergency services	Yes	97	29.1
Difficulty buying the medicines	Yes	81	24.3
	Mean	Median	Min./Max.
Medical tests in the last year	1.5	1.0	0/12
Pharmacy visits (last 3 months)	4.8	3.0	0/36
Physician visits last year	4.9	4.0	1/31

**Table 2 ijerph-19-06170-t002:** Modifiable CVD risk factors of the patients.

Variables	Description	*n*	%
Smoking status	Non-smoker	297	89.2
	Ex-smoker (<5 years)	10	3.0
	Smoker	26	7.8
Diet (vegetables/fruit)	Never	0	0.0
	Sometimes	61	18.3
	Every day	270	81.1
	(Missing)	2	0.6
	≥5 servings/day	89	26.7
Sedentary behavior	No	111	33.3
	Yes (Missing)	221 1	66.4 0.3
Alcohol consumption	No	147	44.1
	Yes	162	48.7
	>30 g/day for M or 20 g/day for F	23	6.9
	(Missing)	1	0.3
Anxiety/Depression	No	124	37.2
	Moderate	166	49.9
	Extreme	40	12.0
	(Missing)	3	0.9
Isolation	Living alone	60	18.0
Dyslipidemia	Yes	235	70.6
	Total cholesterol > 190 mg/dL	98	29.4
	LDL-C (>55, 70, 100, and 116 mg/dL, for very high-, high-, moderate, and low-risk)	217	65.2
	Non-HDL-C (>85,100, and 130 mg/dL, for very-high-, high-, and moderate-risk)	215	64.6
	HDL-C < 40 mg/dL for M or < 46 mg/dL for F	76	22.8
	Triglycerides > 150 mg/dL	118	35.4
Obesity	Overweight: BMI 25–29.9 kg/m^2^	149	44.7
	Obesity: BMI ≥ 30 kg/m^2^	103	30.9
	Waist circ. > 102 cm for M or > 88 cm for F	195	58.6
Hypertension	Yes	233	70.0
	High normal	74	31.8
	Grade 1 hypertension	50	21.5
	Grade 2 hypertension	25	10.7
	Grade 3 hypertension	8	3.4
	Isolated systolic hypertension	76	32.6
Fasting glucose levels	102–125 mg/dL	115	34.5
	≥126 mg/dL	44	13.2
HbA1c	≥6.5	50	15.0

Abbreviation: F—female; M—male.

**Table 3 ijerph-19-06170-t003:** Analysis of the main quality indicators of guideline adherence for the CVD risk factors.

QI		Number of Eligible Cases	Guideline Adherence		Guideline- Adherent	Non-Guideline-Adherent	*p*-Value *
			*n*	%	Controlled/ Non-Controlled	Controlled/ Non-Controlled	
1 ^a^	Patients with hypertension on ACEI or ARB.	229	190	83.0	45/145	9/30	*p* = 0.935
2 ^a^	Patients with hypertension on multiple drug therapy with a RAAS inhibitor, a CCB, and diuretics.	229	54	23.6	17/37	37/138	*p* = 0.118
3 ^b^	Patients with type 2 DM without previous ASCVD, CKD, HF, or with ASCVD, on Metformin, unless contraindicated.	65	57	87.7	43/14	5/3	*p* = 0.436
4 ^a^	Patients with type 2 DM and ASCVD or at very high/high CVD risk on a GLP-1RA or SGLT2i.	67	4	6.0	1/3	49/14	-
5 ^a^	High-intensity statin is prescribed up to the highest tolerated dose and if the LDL-C goals are not achieved.	202	5	2.5	3/2	74/118	*p* = 0.578
6	In smokers, follow-up support, NRT, varenicline, and bupropion individually/in combination should be considered	26	0	0	0/0	0/26	-

^a^ Class of recommendation I, Level of evidence A; ^b^ Class of recommendation I, Level of evidence B; * chi-square test. Abbreviation: ACEI—angiotensin converting enzyme inhibitors; ARB—angiotensin receptor blockers; ASCVD—atherosclerotic cardiovascular disease; CCB—calcium channel blockers; DM—diabetes mellitus; CKD—chronic kidney disease; GLP-1RA—glucagon-like peptide-1 receptor agonists; HF —heart failure; LDL-C—low-density lipoprotein-cholesterol; NRT—nicotine replacement therapy; RAAS—renin-angiotensin-aldosterone system; SGLT2i—sodium-glucose linked transporter 2 inhibitors.

## Data Availability

The data presented in this study are available on request from the first author.

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
