# Peer review of "Evaluation of Cardiovascular Pharmacotherapy Guideline Adherence and Risk Factor Control in Portuguese Community Pharmacy Patients"

_ijerph, 2022, doi:10.3390/ijerph19106170_

Round 1
Reviewer 1 Report
Row 86-92: it is not obvious, that the mentioned investigations are pharmaceutical competencies (in a community pharmacy; independently of the training provided by WCD), this should be explained in more detail.
Row 73-75: there is a contradiction between the two statements.
Row 121: “4th” instead of “4th”.
Table 2: at some categories, e.g. anxiety/depression, diet (vegetables/fruit), sedentary behavior and alcohol consumption the total number (n) in the different categories is more or less than 333.
Row 222: reference regarding the classification of statin-dosage intensity should be included.
Rows 210 and 228: regarding the patients who showed raised HbA1c-level, the n=10 means 3.0% or 3.8%. This difference should be explained.
The newest (2021) version of the ESC guideline should be mentioned and cited in the manuscript (https://academic.oup.com/eurjpc/article/29/1/5/6374862?login=false) and, if there are significant and relevant differences compared to the older version, these should be discussed.
Author Response
Dear Prof. Dr. Paul B. Tchounwou,
Please find below, the point-by-point response to the Reviewer 1 comments and concerns.
***
Reviewer 1,
Row 86-92: it is not obvious, that the mentioned investigations are pharmaceutical competencies (in a community pharmacy; independently of the training provided by WCD), this should be explained in more detail.
RE: We have added the sentence – “In the community pharmacy, these evaluations were performed by a licensed pharmacist. The devices and the training of the investigator pharmacist on equipment operation were provided by World Care & Diagnostics.”
Row 73-75: there is a contradiction between the two statements.
RE: We have made the change in the whole paragraph to better clarify the recruitment process. The new paragraph reads as follows: “The enrollment process occurred in a community pharmacy in a central Portuguese city. All the customers who entered the pharmacy during a 70-hour working week (Monday through Saturday) were registered with name, date of birth, and telephone number. Then, the pharmacist contacted each pharmacy customer over the phone and made an invitation to participate in the study. The pharmacist performed a CVD risk assessment for the participants who agreed to participate in the study in a dedicated counselling room, where privacy could be maintained within the pharmacy with no interference by routine pharmacy processes, for seven months. The inclusion criteria for this study were customers who were on cardiovascular pharmacotherapy. The customers who were not on cardiovascular pharmacotherapy were included in a screening program [13].”
Row 121: “4th” instead of “4th”.
RE: Corrected. Thank you.
Table 2: at some categories, e.g. anxiety/depression, diet (vegetables/fruit), sedentary behavior and alcohol consumption the total number (n) in the different categories is more or less than 333.
RE: In fact, we had missing data in the responses to the question about anxiety/depression (n=3), diet (vegetables/fruit) (n=2), sedentary behavior and alcohol consumption (n=1). We added the missing data to Table 2. Thank you for perceiving this aspect, which, when corrected, improves the quality of the results. The categories “diet (vegetables/fruit)” had as answer “Never” (n=0), “Sometimes” (n=61), and “Every day” (n=270). In a different question we asked about the number of servings per day, and 89 had ≥ 5 servings/ day. We corrected Table 2 and this last category was separated, for the sake of clarity.
Row 222: reference regarding the classification of statin-dosage intensity should be included.
RE: We added the following sentence to the Methods section: “The lipid-lowering therapy with statins was classified into three different dosage-intensity categories, high-, moderate-, and low-intensity statin, according to the American College of Cardiology and the American Heart Association guidelines.” And added the respective bibliography reference [20].
Rows 210 and 228: regarding the patients who showed raised HbA1c-level, the n=10 means 3.0% or 3.8%. This difference should be explained.
RE: We added the considered “n” to each result for better clarification:
Row 229 – “Apart from these CVD risk factors and after excluding the patients with diagnosed type 2 and type 1 diabetes (n=70), among the remaining 263 patients, 33.1% (n=87) patients had fasting glucose levels 102-125 mg/dL, 3.0% (n=8) patients had fasting glucose levels ≥ 126mg/dL, and 3.8% (n=10) patients had HbA1c ≥ 6.5%.
AND
Row 246 - “Of the 333 patients analyzed, 9.6% (n=32) showed high BP, but were not treated for hypertension; 29.1% (n=97) showed high lipid levels, but were not treated for dyslipidemia; and 3.0% (n=10) showed high HbA1c, but were not treated for type 2 diabetes,…”
The newest (2021) version of the ESC guideline should be mentioned and cited in the manuscript (https://academic.oup.com/eurjpc/article/29/1/5/6374862?login=false) and, if there are significant and relevant differences compared to the older version, these should be discussed.
RE: We had mentioned and cited the newest version of the ESC guideline, but did not discuss the differences, which we added now: “The 2021 ESC Guidelines on CVD prevention in clinical practice were released in September 2021 [21]. In terms of target levels for the risk factors, the newest guidelines are in line with the previous guidelines on diabetes and pre-diabetes (2019) [15] on dyslipidemias [16], and on arterial hypertension (2018) [18]. Another relevant difference are the new SCORE2 and SCORE2-OP charts, which predict the 10-year risk of fatal and non-fatal cardiovascular events, while the previous charts predicted the 10-year risk of fa-tal cardiovascular events. These new charts were calibrated to four clusters of countries (low, moderate, high, and very high CVD risk) that were grouped based on national CVD mortality rates published by the World Health Organization. Based on these mortality rates, Portugal was considered as moderate risk region for CVD, which influenced the risk category stratification. Another difference compared to former guidelines is the stepwise approach to risk factor treatment and treatment intensification to reach risk factor goals. This approach is facilitated by the communication to patients of treatment benefits of risk factors in an understandable way, with charts showing the average lifetime benefit of smoking cessation, lipid-lowering, and BP-lowering, expressed as extra CVD-free life-years, which may improve the shared decision-making process.”

Reviewer 2 Report
Very good material for further studies.
In the conclusion - I would suggest to include a proposal of actions for the pharmacist to improve the situation described.
Author Response
Dear Prof. Dr. Paul B. Tchounwou,
Please find below, the point-by-point response to the Reviewer 2 comments and concerns.
***
Reviewer 2,
Very good material for further studies.
In the conclusion - I would suggest to include a proposal of actions for the pharmacist to improve the situation described.
RE: Thank you so much! In fact, this study was also intended to provide a basis for future studies, and the inclusion of this purpose is essential to be stated in the conclusion. Thus, we added the following sentence to the conclusion: “This study may provide the groundwork for further, larger, multicentric studies, to prove the positive impact of pharmacist-led medication review on CVD.”

Reviewer 3 Report
This is an interesting article and potentially shows the roles pharmacists could assume in improving patient outcomes in CVD. The Introduction needs some background as to the availability and scope of CVD services normally provided in Portuguese community pharmacies and whether these are remunerated. The paper contains several grammatical errors and may need assistance of an editor. The Methods are unclear. It says recruitment occurred in a community pharmacy, but later says they were recruited later by telephone? It is unclear why these patients came to the pharmacy. Did they bring a prescription? Were patients stabilised on these CVD medications? Where pharmacists prescribing and/or supplying medications directly to patients? Was a diagnosis available? Were other factors such as kidney function available. The recruitment for 70 hours per working week and over what specific period is unclear. The Results do not indicate the time period of recruitment. It seems amazing that of 1261 customers that entered the pharmacy 1101 were contacted that would potentially fulfill the study criteria. How were the diagnoses confirmed? There were 26 smokers in the Table and 27 in the text (Line 239). Was a patient's history available? Was patient preference for medication considered and patient medication adherence evaluated? High doses of statins for example can lead to side effects impairing adherence to the dosage regimen and some over 80 years, may not benefit from a statin. The study appears to be from one community pharmacy which may limit generalisability. If medication was prescribed how many different prescriber's were involved? This also limits generalisability.
Author Response
Dear Prof. Dr. Paul B. Tchounwou,
Please find below, the point-by-point response to the Reviewer 3 comments and concerns.
***
Reviewer 3,
This is an interesting article and potentially shows the roles pharmacists could assume in improving patient outcomes in CVD.
RE: Thank you so much for pointing out the potential of our study.
The Introduction needs some background as to the availability and scope of CVD services normally provided in Portuguese community pharmacies and whether these are remunerated.
RE: We added the following paragraph to the introduction: “The Portuguese community pharmacist’s scope of practice was expanded in 2018 to include new services routinely provided and enforced by law: nutrition appointments; therapy adherence programs, medicine reconciliation, services utilizing multicompartment aids, and education programs on the use of medical devices; performance of rapid tests for HIV, HCV, and HBV screening (point-of-care tests), including pre-and posttest counselling and referral of positive cases to hospital care; and basic nursing services [11]. Pharmacies may also promote campaigns and programs for health literacy, disease prevention, and healthy lifestyle promotion. Most Portuguese pharmacies perform point-of-care tests that enable CVD risk evaluation. These services are freely priced by pharmacies and paid out-of-pocket by users, but there are no publicly available data on the number of services or pricing [12].”
The paper contains several grammatical errors and may need assistance of an editor.
RE: The manuscript has been revised to correct the errors.
The Methods are unclear. It says recruitment occurred in a community pharmacy, but later says they were recruited later by telephone? It is unclear why these patients came to the pharmacy. Did they bring a prescription?
RE: We have made the change in the whole paragraph to better clarify the recruitment process. The new paragraph reads as follows: “The enrollment process occurred in a community pharmacy in a central Portuguese city. All the customers who entered the pharmacy during a 70-hour working week (Monday through Saturday) were registered with name, date of birth, and telephone number. Then, the pharmacist contacted each pharmacy customer over the phone and made an invitation to participate in the study. The pharmacist performed a CVD risk assessment for the participants who agreed to participate in the study in a dedicated counselling room, where privacy could be maintained within the pharmacy with no interference by routine pharmacy processes, for seven months. The inclusion criteria for this study were customers who were on cardiovascular pharmacotherapy. The customers who were not on cardiovascular pharmacotherapy were included in a screening program [13].”
Where pharmacists prescribing and/or supplying medications directly to patients?
RE: In Portugal the pharmacists dispense and provide over-the-counter (OTC) medication counselling. However, pharmacists are not allowed to prescribe cardiovascular medication, these are supplied only on prescription.
The recruitment for 70 hours per working week and over what specific period is unclear.
RE: The enrolment process occurred only in one single week, from Monday to Saturday with the pharmacy opening hours, from 8:00 am to 8:00 pm from Monday to Friday and from 8:00 am to 18:00 pm on Saturday, totalling the referred 70 hours. A sentence was added: “All the customers who entered the pharmacy during a 70-hour working week (Monday through Saturday)...”
The Results do not indicate the time period of recruitment.
RE: We had indicated the study period in the Method section.
It seems amazing that of 1261 customers that entered the pharmacy 1101 were contacted that would potentially fulfill the study criteria.
RE: We did not know in advance the characteristics of the users who would enter the pharmacy and if they were going to give their phone numbers to be contacted (even though some were already regular customers); what they were going to do in the pharmacy; how many of those would accept to assess the cardiovascular risk by the pharmacist; and how many were actually going to attend the evaluation. Contacting them for the study was a very enriching and interesting experience and we were indeed surprised with the obtained participation rate and the good acceptability of pharmacy customers.
There were 26 smokers in the Table and 27 in the text (Line 239).
RE: Sorry for this typo. Indeed, we had 26 smokers in the patients’ sample and not 27.
Was patient preference for medication considered and patient medication adherence evaluated?
RE: Since the aim of this study was to evaluate the cardiovascular pharmacotherapy guideline adherence, measuring preference for medication and patient medication adherence were out of our scope.
High doses of statins for example can lead to side effects impairing adherence to the dosage regimen and some over 80 years, may not benefit from a statin.
RE: We totally agree with your comment. And we took that into account, as stated in Line 158 to 162 – “Regarding guideline adherence in patients with dyslipidemia we considered the recommendation that a high-intensity statin should be prescribed up to the highest tolerated dose to reach the LDL-C goals set for the specific risk category and if the goals are not achieved with the maximum tolerated dose of a statin, combination with ezetimibe is recommended (QI-5)”. We are also aware that the evidence supporting more strict targets for very old people (>80 years) and those who are frail is not strong, as stated by the ESC guidelines, and that more evidence is required to support the use of statin-based treatment in older people, particularly in those aged ≥80 years. We have taken into account these considerations for very old and frail patients.
Were patients stabilised on these CVD medications? Was a diagnosis available? Were other factors such as kidney function available? How were the diagnoses confirmed? Was a patient's history available?
RE: We explain in the limitations that Portuguese community pharmacists have no access to patients’ medical records, which happens also with all the private healthcare providers (including private physicians). So, all these professionals (i.e., pharmacists and private physicians) must support their clinical practice on thorough patients interviews and anamnesis. We minimized the recall bias asking the patients to bring all the documentation they had available, to support the pharmacist’s interview. It is also important to bear in mind that 84.7% patients were loyal customers of the pharmacy, which facilitates the access to in-pharmacy medication history data.
The study appears to be from one community pharmacy which may limit generalisability. If medication was prescribed how many different prescriber's were involved? This also limits generalisability.
RE: We added as a potential limitation for the study the fact that it was a single-center study. However, nothing made us consider that the population attended in the community pharmacy presents any substantial differences with CVD patients in Portugal.

Round 2
Reviewer 3 Report
The revised version has addressed the comments made, rendering the paper and its context much improved. A related issue that has not been raised is that of patient adherence to their medication regimens. Sub-optimal patient adherence could explain some of the poor outcomes, especially when prescribing was guideline adherent. Patient adherence should be evaluated by pharmacists routinely. Something should be added to the limitations about this likelihood affecting patient outcomes, including when prescribing is guideline adherent. Presumably community pharmacists in Portugal assess this on a routine basis? There are some noticeable grammatical errors.
Line 50: Insert "have" after studies.
Line 52: Change not to non-
Line: 145: Replace "About" with For.
Author Response
The revised version has addressed the comments made, rendering the paper and its context much improved.
RE: Thank you very much, once again, for the time you devoted to help us improving the manuscript.
A related issue that has not been raised is that of patient adherence to their medication regimens. Sub-optimal patient adherence could explain some of the poor outcomes, especially when prescribing was guideline adherent. Patient adherence should be evaluated by pharmacists routinely. Something should be added to the limitations about this likelihood affecting patient outcomes, including when prescribing is guideline adherent. Presumably community pharmacists in Portugal assess this on a routine basis? There are some noticeable grammatical errors.
RE: We agree that poor medication adherence could be one of the several causes of poor patient outcomes. But we must bear in mind that other factors, not easily evaluated at large scale in routine community pharmacy practice, can also contribute to the poor outcomes, namely poor diet or inadequate lifestyles. We added the limitation, as requested.
Line 50: Insert "have" after studies.
RE: Done
Line 52: Change not to non-
RE: Done
Line: 145: Replace "About" with For.
RE: Done